# Manifold Learning via Foliations, and Knowledge Transfer

## Abstract

Understanding how real data is distributed in high dimensional spaces is the key to many tasks in machine learning. We want to provide a natural geometric structure on the space of data employing a deep ReLU neural network trained as a classifier. Through the data information matrix (DIM), a variation of the Fisher information matrix, the model will discern a singular foliation structure on the space of data. We show that the singular points of such foliation are contained in a measure zero set, and that a local regular foliation exists almost everywhere. Experiments show that the data is correlated with leaves of such foliation. Moreover we show the potential of our approach for knowledge transfer by analyzing the spectrum of the DIM to measure distances between datasets.

## 1 Introduction

The concept of manifold learning lies at the very heart of the dimensionality reduction question, and it is based on the assumption that we have a natural Riemannian manifold structure on the space of data (Tenenbaum, 1997; Hinton & Roweis, 2002; Tenenbaum et al., 2000; Fefferman et al., 2013). Indeed, with such assumption, many geometrical tools become readily available for machine learning questions, especially in connection with the problem of knowledge transfer (Bozinovski, 2020; Weiss et al., 2016), as geodesics, connections, Ricci curvature and similar (Ache & Warren, 2019). In particular, the fast developing field of Information Geometry (Amari, 1998; Martens, 2020), is now providing with the techniques to correctly address such questions (Sun & Marchand-Maillet, 2015).

However, the practical situation, arising for example in classifying benchmark datasets as MNIST (Lecun et al., 1998), Fashion-MNIST (Xiao et al., 2017) and similar, with CNNs (Convolutional Neural Networks), is generally more complicated and does not allow for such simple description, at least in the cases most relevant for concrete applications, and calls for more sophisticated mathematical modeling, that we shall explore here replacing the notion of manifold with that of singular foliation. Deep Learning vision classifier models, via their intrinsic hierarchical structure, offer a natural representation and implicit organization of the input data (Olah et al., 2017). For example, Hinton & Salakhutdinov (2006) show how a multilayer encoder network can be successfully employed to transform high dimensional data into low dimensional code, then to retrieve it via a "decoder".

In this paper, we want to provide the data space of a given dataset with a natural geometrical structure, and then employ such structure to extract key information. We will employ a suitably trained CNN model to discern a *foliation* structure in the data space and show experimentally how the dataset our model has been trained with is strongly correlated with its leaves, which are submanifolds of the data space. The mathematical idea of foliation is quite old (see (Reeb, 1961; Ehresmann, 1963) and refs therein); however, its applications in control theory (see (Agrachev & Sachkov, 2004) and refs therein) via sub-Riemannian geometry and machine learning have only recently become increasingly important (Tron et al., 2022; Grementieri & Fioresi, 2022).

The foliation structure on the data space, discerned by our model, however, is non-standard and presents *singular* points. These are points admitting a neighbourhood where the rank of the distribution, tangent at each point to a leaf of the foliation, changes. Moreover, in the presence of typical non linearities of the network, as ReLU, there are also non smooth points, thus hindering in such points the smooth manifold structure itself of the leaf. However, we prove that both singular and smooth points are a measure zero set in the data space, so that the foliation is almost everywhere regular and its distribution well defined. As it turns out in our experiments, the samples belonging to the

dataset we train our model with are averagely close to the set of singular points. It forces us to model the data space with a singular foliation that we call *data foliation* in analogy with the data manifold. Applications of singular foliations were introduced in connection with control theory (Sussmann, 1973); their study is currently an active area of investigation (Lavau, 2018). As we shall show in our experiments, together with their distributions, singular foliations provide with an effective tool to single out the samples belonging to the datasets the model was trained with and at the same time discern a notion of distance between different datasets belonging to the same data space.

Our paper is organized as follows. In Sec. 2, we recap the previous literature, closely related to our work. In Sec. 3, we start by recalling some known facts about Information Geometry in 3.1. Then, in Sec. 3.2 we introduce the data information matrix (DIM), defined via a given model, and two distributions $\mathcal{D}$ and $\mathcal{D}^\perp$ naturally associated with it, together with some illustrative examples elucidating the associated foliations and their leaves. In Sec. 3.3 we introduce singular distributions and foliations and then we prove our main theoretical results expressed in Lemma 3.4 and Theorem 3.6. Lemma 3.4 studies the singularities of the distribution $\mathcal{D}$. Theorem 3.6 establishes that the singular points for $\mathcal{D}$ are a measure zero set in the data space. Hence our foliation, though singular, acquires geometric significance. Finally in Sec. 4 we elucidate our main results with experiments. Moreover, we show how the foliation structure and its singularities can be exploited to determine which dataset the model was trained on, and its distance from similar datasets in the same data space. We also make some "proof of concept" considerations regarding knowledge transfer to show the potential of the mathematical singular local foliation and distribution structures in data space.

## 2  RELATED WORKS

The use of Deep Neural Networks as tools for both manifold learning and knowledge transfer has been extensively investigated. We give a brief overview on the literature closer to our contribution.

**Latent manifold**. The question of finding a low dimensional manifold structure (*latent manifold*) into high dimensional dataset spaces starts with PCA and similar methods to reach the more sophisticated techniques as in (Fefferman et al., 2013; Sun & Marchand-Maillet, 2015), (see also (Murphy, 2022) for a description of the most popular techniques in dimensionality reduction and (Burges, 2010) for a complete bibliography or the origins of the subject). Such understanding was applied towards the knowledge transfer questions in (Cook et al., 2007; Bengio, 2012) and more recently Dikkala et al. (2021). Moreover, distances between datasets were also studied through Optimal Transport in (Alvarez-Melis & Fusi, 2020; Hua et al., 2023). Another important technique of knowledge transfer is via shared parameters as in (Maurer et al., 2016). The idea of using techniques of Information Geometry for machine learning started with Amari (1998). More recently, it was used for manifold learning in (Sun & Marchand-Maillet, 2015) (see also refs therein).

**Foliations.** Employing foliations to reduce dimensionality is not per se novel. In (Grementieri & Fioresi, 2022), the authors introduce some foliation structure in the data space of a neural network, with the aim of approaching dimensionality reduction exploiting the low rank of the equivalent of the Fisher matrix on the dataspace. Observations on the low rank of the Fisher matrix, and its key role in CNN parameter space, appear also in (Sun & Nielsen, 2017; 2024). In (Tron et al., 2022), orthogonal foliations in the data space are used to create adversarial attacks, and provide the data space with curvature and other Riemannian notions to analyse such attacks. In (Szalai, 2023) invariant foliations are employed to produce a reduced order model for autoencoders. Also, in considering singular points for our foliation, we are led to study the singularities of a neural network. This was investigated, for the different purpose of network expressibility in (Hanin & Rolnick, 2019b). The partitioning of the data space into connected components after removing non-smooth points of ReLU CNNs has been studied in (Hanin & Rolnick, 2019a) and in (Montúfar et al., 2014).

## 3  METHODOLOGY

### 3.1  INFORMATION GEOMETRY

The statistical simplex consists of all probability distributions in the form $p(y|x,w) = (p_1(y|x,w), \ldots, p_c(y|x,w))$, where $x$ is a *data point*, belonging to a certain dataset in $\mathbf{R}^d$, divided into $c$ classes, while $w \in \mathbf{R}^N$ are the learning parameters (Amari, 1998; Jost, 2011; Martens,

2020). We define the *information loss* as: $I(x, w) = -\log(p(y|x, w))$ and the *Fisher information matrix* (FIM) (Rao, 1992) as

$$F(x, w) = \mathbf{E}_{y \sim p}[\nabla_w \log p(y|x, w) \cdot (\nabla_w \log p(y|x, w))^T] \quad (1)$$

In analogy to (1) we define the *Data information matrix* (DIM):

$$D(x, w) = \mathbf{E}_{y \sim p}[\nabla_x \log p(y|x, w) \cdot (\nabla_x \log p(y|x, w))^T] \quad (2)$$

As noted in (Sun & Marchand-Maillet, 2015), some directions in the parameter or data space, may be more relevant than others in manifold learning theory. We have the following result (Grementieri & Fioresi, 2022), obtained with a direct calculation.

**Proposition 3.1.** *The Fisher information matrix $F(x, w)$ and the data information matrix $D(x, w)$ are positive semidefinite symmetric matrices. Moreover*[1]:

$$\ker F(x, w) = (\mathrm{span}_{i=1,\ldots,c}\{\nabla_w \log p_i(y|x, w)\})^\perp,$$
$$\ker D(x, w) = (\mathrm{span}_{i=1,\ldots,c}\{\nabla_x \log p_i(y|x, w)\})^\perp. \quad (3)$$

*where the orthogonal $\perp$ is taken with respect to the euclidean product. In particular* $\mathrm{rank}\, F(x, w) < c$ *and* $\mathrm{rank}\, D(x, w) < c$.

Notice that $F(x, w)$ is a $N \times N$ matrix, while $D(x, w)$ is a $d \times d$ matrix, where in typical applications (e.g. classification tasks) $N, d >> c$. Hence, both $F(x, w)$ and $D(x, w)$ are, in general, singular matrices, with ranks typically low with respect to their sizes, hence neither $F(x, w)$ nor $D(x, w)$ can provide meaningful metrics respectively on parameter and data spaces. We shall now focus on the data space.

The result (3) suggests to consider the distribution $\mathcal{D}$:

$$\mathbf{R}^d \ni x \mapsto \mathcal{D}_x := \mathrm{span}_{i=1,\ldots,c}\{\nabla_x \log p_i(y|x, w)\} \subset T_x \mathbf{R}^d \quad (4)$$

where we assume the learning parameters $w \in \mathbf{R}^N$ to be fixed as $x$ varies in the dataspace. In general a distribution on a manifold $M$ assigns to each point a subspace of the tangent space to $M$ at that point (see (Tu, 2011; 2017) for more details), in our case $M = \mathbf{R}^d$.

**Observation 3.1.** The distribution $\mathcal{D}$ at each point in the dataspace coincides with the image of $D$ at that point as defined in (2). This is a direct check, (see (Grementieri & Fioresi, 2022)). Hence the rank of $\mathcal{D}$ at $x$, i.e. the dimension of $\mathcal{D}_x$, and the rank of $D$ at x, i.e. the dimension of its image, coincide.

## 3.2 DISTRIBUTIONS AND FOLIATIONS

Whenever we have a distribution on a manifold $M$, it is natural to ask whether it is *integrable*. A distribution $\mathcal{C}$ on $M$ is *integrable* if for every $x \in M$, there exists a connected immersed submanifold $N$ of $M$, with $T_y N = \mathcal{C}_y$ for all $y \in N$. In other words, the distribution defines at each point the tangent space to a submanifold. Whenever a distribution is integrable, we have a *foliation*, that is, $M$ becomes a disjoint union of embedded submanifolds called the *leaves* of the foliation. If $\dim \mathcal{C}_x = \dim \mathcal{C}_y$ for all $x, y \in M$ we say that the foliation, or the corresponding distribution, has *constant rank*. Fig. 1 illustrates a constant rank foliation and its orthogonal complement foliation, with respect to the euclidean metric in $M = \mathbf{R}^3$ (the ambient space). In this paper, we focus on the study of the distribution $\mathcal{D}$ defined in (4), strictly related with the DIM $D$, as defined in (2), since $\mathcal{D}$ is equivalently spanned by the columns of $D$ (see Obs. 3.1). Under suitable regularity hypothesis, Frobenius theorem (Tu, 2011; 2017) provides with a characterization of integrable distributions.

**Theorem 3.2. (Frobenius Theorem).** *A smooth constant rank distribution $\mathcal{C}$ on a real manifold $M$ is integrable if and only if it is involutive, that is for all vector fields $X, Y \in \mathcal{C}$, $[X, Y] \in \mathcal{C}$.* [2]

---

[1]The span of a set of vectors is the set of all their linear combinations.

[2]If $X$, $Y$ are vector fields on a manifold, we define their Lie bracket as $[X, Y] := XY - YX$.

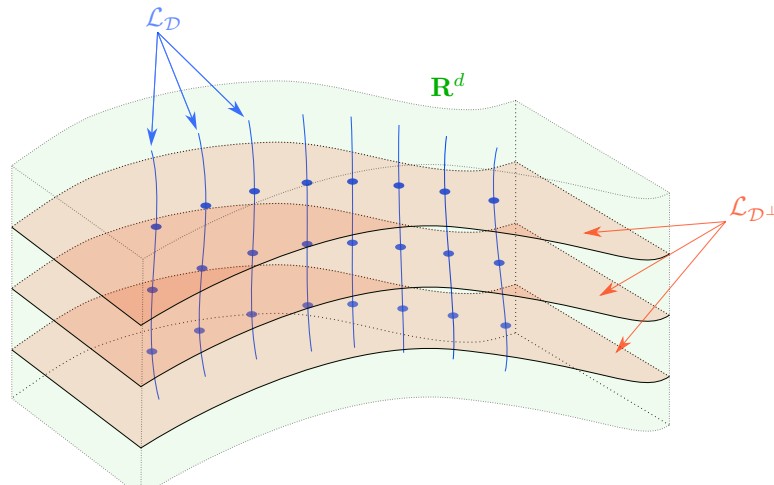

Figure 1: Foliations: $\mathcal{L}_\mathcal{D}$ and $\mathcal{L}_{\mathcal{D}^\perp}$ denote the set of the leaves in distributions $\mathcal{D}$ and $\mathcal{D}^\perp$ respectively.

To elucidate this result in Fig. 4 we look at two examples of foliations of 1-dimensional distributions, obtained from the distribution $\mathcal{D}$ in (4) of a neural network trained on the Xor function, with non linearities GeLU and ReLU[3]. Notice that since in both cases $\mathcal{D}$ is 1-dimensional, then it is automatically integrable (bracket of vector fields being zero). In a more general setting, especially relevant for the applications, we have the following result; the proof is a simple calculation based on the Frobenius Theorem (Grementieri & Fioresi, 2022).

**Proposition 3.3.** *Let the notation be as above. For an empirical probability $p$ defined via a deep ReLU neural network the distribution $\mathcal{D}$ in the data space:*

$$x \mapsto \mathcal{D}_x = \mathrm{span}_{i=1,\ldots,c}\{\nabla_x \log p_i(y|x,w)\}$$

*is locally integrable at smooth points.*

We call the foliation resulting from Prop. 3.3 the *data foliation*. Its significance is clarified by the following examples.

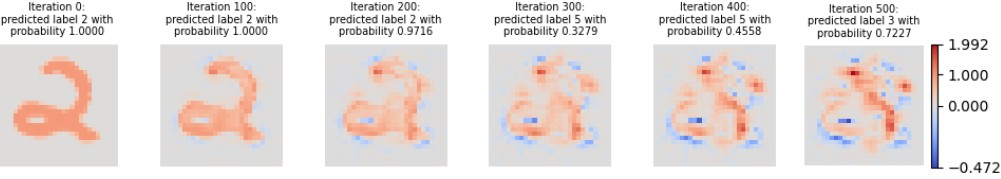

Figure 2: Moving tangentially to $\mathcal{D}$ i.e. in a direction contained in $\mathcal{D}$.

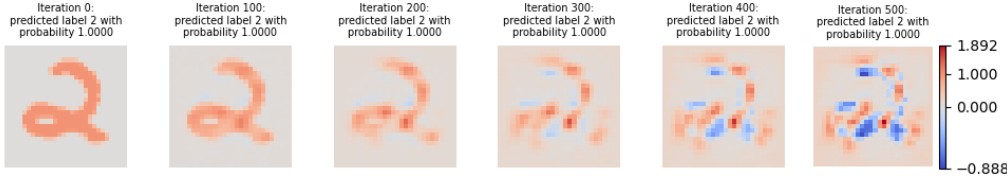

Figure 3: Moving tangentially to $\mathcal{D}^\perp$.

---

[3]We look only at smooth points, which form an open set.

As an illustration of the distributions $\mathcal{D}$ and $\mathcal{D}^\perp$ (orthogonal computed with respect to the euclidean metric in $M = \mathbf{R}^d$), in MNIST, we present Fig. 2 and Fig. 3. We notice that, while moving tangentially to $\mathcal{D}$ from one data point, the model predicts a meaningful label, while moving tangentially to $\mathcal{D}^\perp$, the model maintains high confidence on the label, though not in line with the image. Both Fig. 2 and Fig. 3 were obtained by projecting the *same* direction on $\mathcal{D}$ and $\mathcal{D}^\perp$ at each step. Notice that the phenomena we observe above are due to the fact that the neural network output probabilities are invariant by moving in the kernel of the DIM, that is the distribution $\mathcal{D}^\perp$.

For a model with GeLU non linearity however, one can see experimentally that we do not have the involutivity property anymore for the distribution $\mathcal{D}$ (see Table 1). Hence, there is no foliation whose leaves fill the data space, naturally associated to $\mathcal{D}$ via Frobenius Theorem. To see this more in detail, we define the space $\mathcal{V}_x^D$ generated by $\mathcal{D}_x$ and the Lie brackets of their generators:

$$\mathcal{V}_x^D = \mathrm{span}\{\nabla_x \log p_i(y|x,w), [\nabla_x \log p_j(y|x,w), \nabla_x \log p_k(y|x,w)], i,j,k = 1,\ldots,c\} \tag{5}$$

In Table 1 we report averages of the dimensions of the spaces $\mathcal{D}_x$, $\mathcal{V}_x^D$ for a sample of 100 points

Table 1: Involutivity of the distribution $\mathcal{D}$

| Non linearity | dim $\mathcal{D}_x$ | dim $\mathcal{V}_x^D$ |
|---|---|---|
| ReLU | 9 | 9 |
| GeLU | 9 | 44.84 |
| Sigmoid | 9 | 45 |

$x \in \mathbf{R}^d$. The non involutivity of the distribution is deduced from the fact the dimension increases when we take the space spanned by the distribution and the brackets of its generators. As we can see, while for the ReLU non linearity $\mathcal{D}$ is involutive and we can define the data foliation, the brackets of vector fields generating $\mathcal{D}$ do not lie in $\mathcal{D}$ for the GeLU and sigmoid non linearities. Consequently, there is no foliation and the sub Riemannian formalism appears more suitable to describe the geometry in this case. We shall not address the question here.

### 3.3 Singular Foliations

A foliation on a manifold $M$ is a partition of $M$ into connected immersed submanifolds, called leaves. A foliation is *regular* if the leaves have the same dimension, *singular* otherwise. Notice that the map $x \mapsto \dim(\mathcal{L}_x)$, which associates to $x \in M$ the dimension of its leaf $\mathcal{L}_x$, is lower semi-continuous, that is, the dimensions of the leaves in a neighbourhood of $x$ are greater than or equal to $\dim(\mathcal{L}_x)$ (Lavau, 2018). Whenever we have an equality, we say that $x$ is *regular*, otherwise we call $x$ *singular*. It is important to remark that, adhering to the literature (Lavau, 2018), the terminology *singular* point here refers to a point that has a neighbourhood where the leaves have non constant dimension. We can associate a distribution to a foliation by associating to each point, the tangent space to the leaf at that point. Such distribution has constant rank if and only if the foliation is regular. Frobenius Theorem 3.2 applies only to the case of constant rank distributions, however, there are results extending part of Frobenius theorem to non constant rank distributions. In (Hermann, 1963) and Nagano (1966), the authors give sufficient conditions for integrability in this more general setting (see (Lavau, 2018) for correct attributions on statements and proofs).

In the practical applications we are interested in, foliations may also have *non smooth* points, that is points where the leaf through the point, belonging to the foliation, is not smooth. For example, Fig. 4 shows a `Xor` network with ReLU non linearity displaying both singular and non smooth points.

The singular point of each picture is in the center, while non smooth points occur only for `Xor` with ReLU non linearity. As previously stated, this is a degenerate case, where the involutivity and integrability of $\mathcal{D}$ is granted automatically by low dimensionality, though for (b) in Fig. 4, we cannot define $\mathcal{D}$ at non smooth points.

In many interesting applications involving neural networks with ReLU non linearity, the distribution $\mathcal{D}$, generated by the image of the DIM $D$, is both non smooth and singular. It then becomes important to understand where such non smooth and singular points are located. The aim of this section is

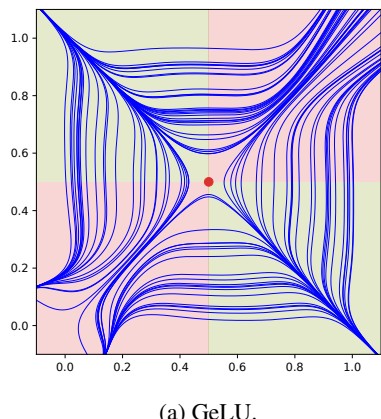

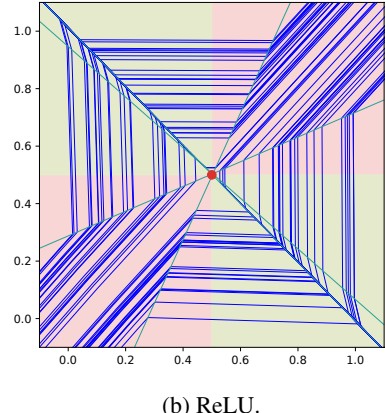

(a) GeLU.                (b) ReLU.

Figure 4: The blue lines are a sample of the data foliation defined by the distribution $\mathcal{D}$ (4) for a `Xor` network. The two classes of the `Xor` problem are represented in red and green squares underneath. The red dot is a singular point for the foliation. In (b), the green lines are the non-smooth points.

to show they occupy a measure zero set in the dataspace, so that we can indeed apply successfully Frobenius theorem in a large portion of the dataspace. In our next section we shall investigate experimentally their location in dataspace and the importance in the knowledge transfer questions.

**Remark 3.2.** Notice that Frobenius Theorem 3.2 and its non constant rank counterparts, apply only in the smoothness hypothesis, while for applications, i.e. the case of ReLU networks, it is necessary to examine also non smooth foliations. We plan to explore the non smooth setting more generally in a forthcoming paper.

We now want to investigate further the singular points of the distribution $\mathcal{D}$.

Let $N(x) = p(y|x, w) = (p_1(y|x, w), \ldots, p_c(y|x, w))$ denote the output of the neural network classifier and $J_p(x) = (\nabla_x p_i(y|x, w))$ its Jacobian matrix. So $\nabla_x p_i(y|x, w)$ is the $i$-th row (column) of $J_p(x)$. One can see, by the very definition of $\mathcal{D}$ (4), that $\mathrm{rank}\mathcal{D} = \mathrm{rank}J_p$. We assume $w$ to be constant, that is we fix our model. Let $P = \mathrm{diagonal}(p_1(y|x, w), \ldots, p_c(y|x, w))$.

From now on we assume $N(x) = p(y|x, w) = \mathrm{Softmax} \circ S(x)$ where $S(x)$ represents the score. Then we have:

$$J_p(x) = (P - p^t p)J_S(x) \tag{6}$$

where $p = (p_1(y|x, w), \ldots, p_c(y|x, w))$ and $J_S(x)$ is the Jacobian of the score.

To study the drop of rank for $\mathcal{D}$ or equivalently for $J_p$, let us first look at the kernel of $(P - p^t p)$.

**Lemma 3.4.** *Let $E_i$ denote the vector with the $i$-th coordinate equal to one, and the others equal to zero.*

$$\ker(P - p^t p) = \mathrm{span}\{(1, \ldots, 1)\} + \mathrm{span}\{E_i, \ \forall i \text{ such that } p_i = 0\} \tag{7}$$

*Proof.* Let $u \in \mathbf{R}^c$. Then $u \in \ker(P - p^t p)$ if and only if $p_i u_i - p_i \sum_k p_k u_k = 0$. This is equivalent to:

$$p_i = 0 \quad \text{or} \quad u_i - \sum_k p_k u_k = 0 \iff p_i = 0 \quad \text{or} \quad \sum_k p_k(u_i - u_k) = 0$$

The inclusion $\supseteq$ is thus straightforward. To get the other inclusion, let $i_0$ denote the argmax of $u$. Then, $\sum_k p_k(u_{i_0} - u_k)$ is a sum of non negative terms; hence to be equal to zero, must be $p_k(u_{i_0} - u_k) = 0$ for all $k$. Therefore, $u_k = u_{i_0}$ for all $k$ such that $p_k \neq 0$. This is enough to prove the direct inclusion $\subseteq$. $\qquad\square$

We have the following important observation, that we shall explore more in detail in our experiments.

**Observation 3.3.** Lemma 3.4 tells us that the rank of the distribution $\mathcal{D}$ or equivalently of $J_p(x) = (P - p^t p)J_S(x)$ is lower at points in the data space where the probability distribution has higher number of $p_i \neq 0$. Clearly the points in the dataset, on which our model is trained, are precisely the

points where the empirical probability $p$ is mostly resembling a mass probability distribution. Hence at such points, we will observe empirically an average drop of the values of the eigenvalues of the DIM (whose columns generate $\mathcal{D}$), compared to random points in the data space, as our experiments confirm in Sec. 4. As we shall see, this property characterizes the points in the dataset the model was trained with.

In our hypotheses, since the probability vector $p$ is given by a Softmax function, it cannot have null coordinates. Therefore, Lemma 3.4 states that $\dim \ker(P - p^t p) = 1$ and that the kernel of $P - p^t p$ does not depend on the input $x$. Thus, the drops in rank of $\mathcal{D}$ does not depend on $\ker(P - p^t p)$ and can only be caused by $J_S(x)$.

Now we assume that the score $S$ is a composition of linear layers and activation functions as follows:

$$S(x) = L_{W_\ell} \circ \sigma \circ \cdots \circ \sigma \circ L_{W_1} \tag{8}$$

where $\sigma$ is the ReLU non linearity, $L_{W_i}$ are linear layers (including bias) and $\ell$ is the total number of linear layers. We denote the output of the $k$-th layer:

$$f_k(x) = L_{W_k} \circ \sigma \circ \cdots \circ \sigma \circ L_{W_1}(x)$$

Let us define, for a subset $U$ in $\mathbf{R}^d$:

$$Z_U = \{x \in U \text{ such that } \exists i, \ x_i = 0\} \tag{9}$$

**Lemma 3.5.** *Let $\mathcal{O}$ be the set of points in $M = \mathbf{R}^d$, admitting a neighbourhood where the score function $S$ is non constant. Then, the set of singular points of $J_S$, the Jacobian of $S$, is a subset of:*

$$\bigcup_{k=1}^{\ell-1} f_k^{-1}(Z_{f_k(M)}) \cap \mathcal{O} = \bigcup_{k=1}^{\ell-1} \bigcup_{i=1}^{\dim f_k(M)} \{x \mid f_k(x)_i = 0\} \cap \mathcal{O} \tag{10}$$

*This set is the finite union of closed null spaces, thus of zero Lebesgue measure.*

*Proof.* A short calculation based on the expression of $S$ (8) gives:

$$J_S(x) \quad = \quad W_\ell J_\sigma\left(f_{\ell-1}(x)\right) W_{\ell-1} J_\sigma\left(f_{\ell-2}(x)\right) \ldots J_\sigma(f_1(x)) W_1 \tag{11}$$

Notice that:

$$(J_{\text{ReLU}}(x))_{i,j} = \begin{cases} \delta_{i,j} & \text{if } x_i > 0 \\ 0 & \text{if } x_i < 0 \end{cases} \tag{12}$$

Hence, the set $Z_U$ represents the singular points of $J_{\text{ReLU}}$ on the domain $U$.

If $x_0 \in \{x \mid f_k(x)_i = 0\} \cap \mathcal{O}$, then, even if it means making infinitesimal changes to the network weights, there exists a neighborhood of $x_0$ such that $(f_k)_i$ is linear on it. Besides, since $x_0 \in \mathcal{O}$, again, even if it means making infinitesimal changes to the network weights, then $(f_k)_i$ will not be trivial. Thus $\{x \mid f_k(x)_i = 0\} \cap \mathcal{O}$ is contained in an hyperplane with dimension $< d = \dim M$. $\square$

Now we see that singular points occur on a (Lebesgue) measure zero set.

**Theorem 3.6.** *Let the notation be as above. Consider the distribution $\mathcal{D}$:*

$$\mathbf{R}^d \ni x \mapsto \mathcal{D}_x = \text{span}\{\nabla_x p_i(y|x, w), \ i = 1, \ldots c\} \tag{13}$$

*where $p$ is an empirical probability given by Sofmax and a score function $S$ consisting of a sequence of linear layers and ReLU activations. Then, its singular points (i.e. points where $\mathcal{D}$ changes its rank) are a closed null subset of $\mathbf{R}^d$ contained in the union of $\ell$ hypersurfaces, where $\ell$ is the number of layers.*

*Proof.* The singular points of $\mathcal{D}$ coincide, by Lemma 3.4 with the points where $J_S$ the Jacobian of $S$ as in (8 changes its rank. By Lemma 3.5 this occurs in the union of $\ell$ hypersurfaces. $\square$

As a consequence of the proofs of Lemma 3.5 and Thm 3.6, the singular points of the distribution $\mathcal{D}$ are contained in the non smooth points and such points are contained in a measure zero subset of the data space. Hence, if we restrict ourselves to the open set complementing such measure zero set, we can apply Frobenius Thm 3.2 to the distribution $\mathcal{D}$ to get a foliation. Since it is of significance in the dataspace, as we are going to elucidate in the next section, we call such foliation the *data foliation*.

**Remark 3.4.** Thm 3.6 holds also in the more general hypothesis of piecewise linear activation functions, in place of ReLU.

**Observation 3.5.** There is empirical evidence (Hanin & Rolnick, 2019a) that, removing the measure zero set of non smooth points as in Thm 3.6, partitions the dataspace into disjoint connected components. In each component the distribution $\mathcal{D}$ has constant rank, hence the foliation restricted to one component is regular. However the global foliation on the dataspace, granted locally by Frobenius Thm 3.2, may vary its rank as we move from one connected component to another, as the experiments in our next section suggest. Hence the notion of singular foliation, i.e. a foliation with leaves of different dimensions, become essential to properly model the dataspace, in which we train the neural network.

## 4 EXPERIMENTS

In this section we look at some experiments to elucidate the role of the singular and non-smooth points (see Thm 3.6) for the distribution $\mathcal{D}$, generated by the image of the DIM $D$ (see Def. (2)), in knowledge transfer questions. The emergence of a singular foliation structure in an dense open set of the dataspace, granted by Thm 3.6, via Frobenius Thm 3.2 (see also Obs. 3.5) is related with the inner structure of datasets in such dataspace. Indeed, we will show that an effective way to measure the "distance" between different datasets embedded in the same dataspace, is related with the rank of $\mathcal{D}$, or equivalently $D$, and the (relative) magnitude of $D$'s eigenvalues.

We perform our experiments on the following datasets: MNIST (Lecun et al., 1998), Fashion-MNIST (Xiao et al., 2017), KMNIST (Clanuwat et al., 2018) and EMNIST (Cohen et al., 2017), letters only, that we denote with Letters. We also create a dataset that we call CIFARMNIST: it is the CIFAR10 dataset (Krizhevsky, 2012) cropped and transformed to be $28 \times 28$ gray-scale pictures.

Our neural network is similar to LeNet, with two convolutional layers, followed by a Maxpool and two linear layers with ReLU activation functions, see Fig. 5. This is slightly more general than our hypotheses in Sec. 3.3. The model is then trained on MNIST, reaching $98\%$ of accuracy.[4]

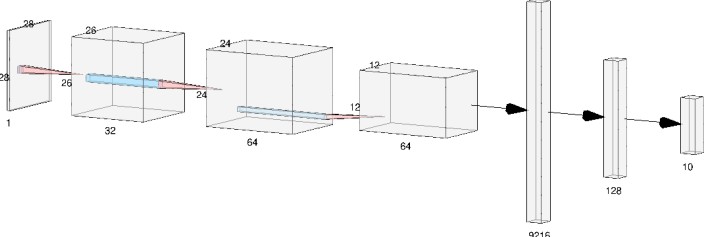

Figure 5: The structure of our CNN - picture created with LeNail (2019).

In Fig. 6 we compute the DIM and we measure its rank by looking at its eigenvalues for 100 sample points in the data space $\mathbf{R}^{784}$ of the MNIST dataset and on 100 uniformly random points in $[0, 1]^{784}$. The statistical significance of these experiments is detailed in Fig. 7. We see clearly that on points in the dataset the eigenvalues of the DIM $D$, are smaller. Since $\mathrm{rank} D = \mathrm{rank} \mathcal{D}$ at each point (see Sec. 3.2), the drop of the eigenvalues of $J_p$ reflects the change in rank of the distribution $\mathcal{D}$ in the proximity of datapoints, i.e. the points on which network was trained. This provides evidence for modeling the dataspace via a singular foliation (see Obs. 3.5. and Lemma 3.4).

We report DIM's eigenvalues $\lambda_{(d)}, \dots, \lambda_{(d-9)}$, sorted in descending order[5] (logarithmic scale), for different datasets in Fig. 7. The vertical segment for each eigenvalue and each dataset represents the values for 80% of the samples, while the colored area represents the values falling in between first and third quartile. The horizontal line represents the median and the triangle represents the mean. The points in MNIST, the training dataset, are clearly identifiable by looking at the colored area.

---

[4]All the code used in this section is available in supplementary material.

[5]Eigenvalues $\lambda_{(d-10)}, \dots, \lambda_{(1)}$ are not plotted because $\mathrm{rank} D < c = 10$, and therefore are equal to zero. $\lambda_{(d-9)}$ is plotted only for reference of what the numerical zero is.

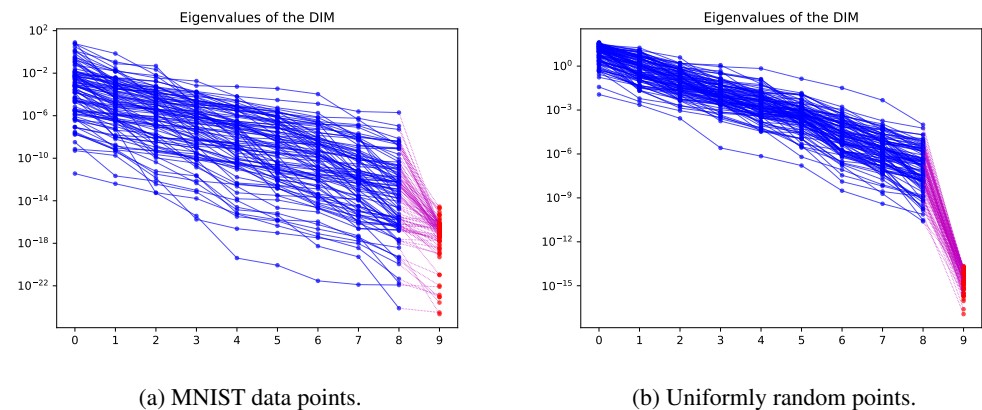

(a) MNIST data points.

(b) Uniformly random points.

Figure 6: DIM eigenvalues sorted by descending order evaluated on 100 points. Each blue line correspond to one data point.

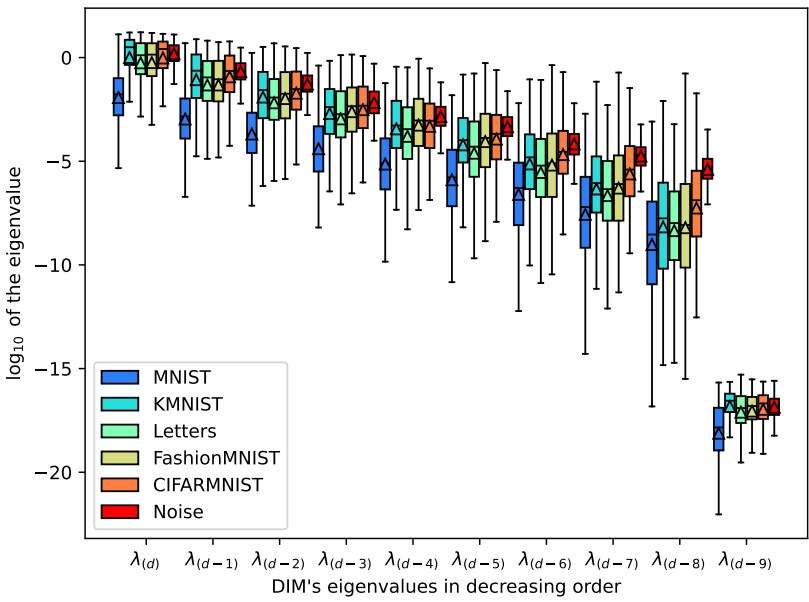

Figure 7: DIM eigenvalues sorted by descending order evaluated on 250 points for each dataset.

We perform a proof of concept knowledge transfer experiment by retraining the last linear layer of our model on different datasets. This experiment was selected as it relies on the degree of similarity between datasets from the perspective of the model. We report in Table 2 the median of highest ($\lambda_{(d)}$) and lowest non-zero ($\lambda_{(d-8)}$) DIM eigenvalues in logarithmic scale, $\Delta$ the average length of the vertical segments in 7, and the validation accuracy after retraining of our CNN. We see a correspondence between the median of the lowest non-zero eigenvalue and the validation accuracy, suggesting to explore in future works the relation between DIM, foliations and knowledge transfer. The significance of the last non-zero eigenvalue of the DIM lies in its correspondence with the rank of the foliation. Indeed, the eigenvalue $\lambda_{(d-8)}$ being (close to) zero means that the rank drops at (close to) the data point. This supports our conjecture that similar datasets (i.e. with high knowledge transfer potential) will lead to data leaves with similar geometric properties.

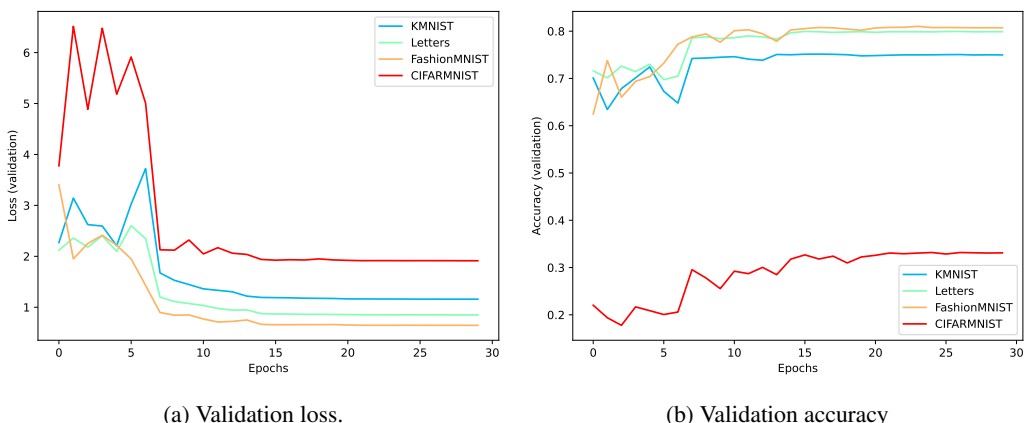

(a) Validation loss.           (b) Validation accuracy

Figure 8: Loss and accuracy after knowledge transfer starting from the weights of a ReLU network trained on MNIST ($98\%$ of accuracy) and retraining only the last linear layer.

Table 2: Parameters for Knowledge Transfer (logarithmic scale)

| Dataset | $\lambda_{(d)}$ | $\lambda_{(d-8)}$ | $\Delta$ | DIM Trace | Val. Acc. |
|---|---|---|---|---|---|
| MNIST | -1.78 | -8.58 | 4.91 | -1.52 | 98% |
| Fashion-MNIST | 0.14 | -8.08 | 4.85 | 0.12 | 81% |
| Letters | 0.11 | -7.99 | 4.75 | 0.48 | 80% |
| KMNIST | 0.49 | -7.75 | 4.74 | 0.37 | 75% |
| CIFARMNIST | 0.41 | -6.90 | 3.84 | 0.27 | 33% |
| Noise | 0.24 | -5.36 | 1.72 | 0.27 | NA |

## 5 CONCLUSIONS

We propose to complement the notion of data manifolds with the more general one of *data foliations*. The (integrable) distribution $\mathcal{D}$, defined via the data information matrix (DIM), a generalization of the Fisher information matrix to the space of data, indeed allows for the partition of the data space according to the leaves of such a foliation via the Frobenius theorem. Examples and experiments show a correlation of data points with the leaves of the foliation: moving according to the distribution $\mathcal{D}$, i.e. along a leaf, the model gives a meaningful label, while moving in the orthogonal directions leads to greater and greater classification errors. The data foliation is however both singular (drop in rank) and non smooth. We prove that singular points are contained into a set of measure zero, hence making the data foliation significant in the data space. We show that points in the dataset the model was trained with have lower DIM eigenvalues, so that the distribution $\mathcal{D}$ allows successfully to determine whether a sample of points belongs or not to the dataset used for training. We make such explicit comparison with similar datasets (i.e. MNIST versus FashionMNIST, KMNIST etc). Then, we use the lowest non-zero eigenvalue of the DIM to measure the distance between data sets. We test our proposed distance by retraining our model on datasets belonging to the same data spaces and checking the validation accuracy. Our results are not quantitatively conclusive in this regard, but they represent a "proof of concept", to encourage further investigation. We believe that our empirical examples show a great promise as a first step to go beyond the manifold hypothesis and exploiting the theory of singular foliations to perform dimensionality reduction and knowledge transfer.

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
