# OpenReview forum: "Manifold Learning via Foliations, and Knowledge Transfer"
_ICLR.cc/2025/Conference — Submitted to ICLR 2025_

### Official Review · Reviewer_qT62 · 2024-11-01

**Soundness:** 3
**Presentation:** 2
**Contribution:** 2
**Rating:** 5
**Confidence:** 2

**Summary:**

This paper presents a geometric analysis of neural networks through the lens of foliation theory. The authors introduced the Data Information Matrix (DIM), a variant of the Fisher Information Matrix applied to data space, to study how neural networks organize high-dimensional data. They developed theoretical proofs showing that neural networks create a foliation structure in the data space, where singular points exist in a measure-zero set. To validate their theory, they conducted experiments using a LeNet-like CNN architecture on MNIST-family datasets, demonstrating how DIM eigenvalues correlate with transfer learning success. They analyzed both ReLU and GeLU networks, showing different properties in terms of foliation structure and investigating the relationship between geometric properties and knowledge transfer capabilities. The experimental validation focused on small-scale problems with 28x28 grayscale images and primarily 10-class classification tasks.

**Strengths:**

### Theoretical Foundation:
The paper provides rigorous mathematical foundation for understanding neural networks geometrically. Authors prove that singular points of the foliation structure lie in a measure-zero set, giving theoretical guarantees for their geometric framework. This connects deep learning to classical differential geometry in a novel way.

### Interpretable Framework:
The authors demonstrate clear visualization of how neural networks organize data through geometric structures. In their XOR example (Figure 4), they show how different nonlinearities (ReLU vs GeLU) create different foliation patterns, making abstract concepts concrete and visually interpretable.

### Quantitative Metrics:
The work introduces measurable quantities for dataset similarity through DIM eigenvalues. Their experiments show that MNIST transfer to Fashion-MNIST achieves 81% accuracy with λ(d-8) = -8.08, while transfer to CIFARMNIST only achieves 33% with λ(d-8) = -6.90, providing a quantitative relationship between geometric properties and transfer learning success.

**Weaknesses:**

### Scale Limitations:
The framework faces severe computational challenges with modern-scale problems. For a typical 224x224x3 image, the DIM would require ~85GB memory, making it impractical for real applications. The experiments only deal with 28x28 grayscale images, avoiding these scaling challenges.

### Dataset Constraints:
The validation is limited to simple classification scenarios. While the theory claims generality, experiments only use MNIST-like datasets with 10 classes. This leaves open questions about handling CIFAR100 (100 classes) and ImageNet (1000 classes).

### Architectural Restrictions:
The theoretical analysis mainly focuses on ReLU networks with simple architectures. Modern networks using complex architectures like ResNet (or VGGNet) are not addressed. For example, their transfer learning experiments only retrain the last layer, which is far from contemporary transfer learning practices that often involve partial fine-tuning or adapter modules.

**Questions:**

I think this paper is trying to contribute theoretically inspired methodology. Thus, would you mind answer the points I mentioned in the weakness?

Please carefully use \citet and \citep in your draft.

---

> ### Author Response · Authors · 2024-11-20
> **Answer to review**
>
> Dear reviewer, we thank you for your comments and questions. To answer the points mentioned in the weakness part:
> 1. Scale Limitations:
> Storing the DIM can indeed be a problem for high-dimensional images on a normal computer. There are a couple of tricks that might alleviate this problem. The first is to never actually compute the DIM but only the Jacobian matrix of the network, since the number of classes is usually much smaller than the dimension of the input. With this Jacobian computed, you can evaluate the FIM on vectors by first computing the product $J_N v$ instead of the product $J_N^t J_N$. This should reduce the overhead quite a bit, depending on your application (in your example, storing the DIM would use ~100Mo with 100 classes). For eigenvalue decomposition, this method coupled with the power iteration method to find eigenvectors is time and memory saving. Finally, if the full DIM is really needed on such large dataset, the use of computing centres is required, which is often already the case for the training phase.
> 2. Dataset Constraints:
> Due to time constraints, and because it is not the main purpose of this paper, we have chosen to perform experiments on reasonable, but sufficiently diverse datasets. We would like to stress that the main goal of this paper is to advertise a geometric framework built to study the knowledge of a trained neural network. The proof of concept given in the experiment section is one in many cases this framework could be used to better understand datasets and ReLU networks, and not there to prove superiority over existing methods.
> 3. Architectural Restrictions:
> As stated in our answer to your second question, we do not aim to perform state of the art knowledge transfer, but rather to use the characteristics of the DIM (here the rank / eigenvalues) to better understand datasets and to what extent they are similar in the eyes of ReLU networks.
>
> We have also corrected the use of \citet and \citep.
> We have highlighted in red the text we modified in the article.

---

### Official Review · Reviewer_L83N · 2024-11-01

**Soundness:** 2
**Presentation:** 1
**Contribution:** 3
**Rating:** 5
**Confidence:** 2

**Summary:**

The submission proposes an information geometry analysis of the data manifold.  In particular, with the “data information matrix” (DIM) -- analogous to the Fisher information matrix but where gradients are taken in data space -- the authors define a distribution over dataspace that can reveal aspects of the data manifold given a trained model.
With this distribution in hand, the authors empirically examine its nature as a foliation (or not, depending on the nonlinear activation in the network) and find that the points in the dataset are different from random points sampled over the dataspace.
From the ability to assess the rank of the distribution at different points in dataspace via the eigenvalues of the DIM, the authors compare the values at points from different datasets for a network trained on MNIST.  Finally, the MNIST-trained network is fine-tuned on the different datasets and the performance of the network is shown to correlate with the smallest non-zero eigenvalues.

**Strengths:**

The combination of differentiable geometry and information geometry to study the data manifold is interesting, and the authors demonstrate that it can lead to some useful insights on real datasets.  In particular, the comparison of the eigenvalue spectrum of the DIM at data points from different datasets seems a reasonable and principled way to assess the model’s familiarity to different datasets.  The DIM seems like a useful construct, and should generally be more manageable to deal with than the FIM (data space being significantly smaller than parameter space); while it was introduced elsewhere, the authors demonstrate new utility from the DIM.

**Weaknesses:**

The text is hard to follow at times.  Given the machine learning audience of ICLR, clarity of exposition about differentiable geometry concepts is important, yet explanations are often confusing or absent (see questions below).  I did not get anything from the visualization of Fig 4 (ReLU vs GeLU trained on XOR).  I am skeptical about the claim that points in the training data are identifiable -- perhaps en masse, but not individually.

**Questions:**

- Why is $D$ used for the dataset in line 113 and then immediately after, for the DIM?
- What are the blue lines showing in Fig 4, and why are they irregularly spaced?  What is the significance of the coloring of points?
- “Hence, there is no foliation whose leaves fill the data space, naturally associated to it via Frobenius Theorem.” (L227)  --> The foliation does not fill data space, but rather the distribution defined by Eqn 4, correct?  What does the “naturally associated...” clause mean, and what does “it” refer to?
- In Figs. 2 and 3, isn’t the distribution as defined in L193 9-dimensional?  So is there an unambiguous sense of direction “moving tangentially to $\mathcal{D}$”?  What is the “same direction” (L212), just a randomly sampled direction in dataspace?
- What is necessary about integrability when making Figs 2 and 3?  Couldn’t similar directions be found in dataspace using the distribution definition (4) for a network with any arbitrary activation function?  What would change about Figs 2 and 3 for a non integrable distribution?
- L306, are p and N(x) equivalent, and if so, why are they both included?
- “The points in MNIST, the training dataset, are clearly identifiable by looking at the colored area.” This seems to be one of the main experimental results.  The overlap between the distributions of eigenvalues suggests only a fairly large population of samples can be identified as the training data or not, right?  Additionally, the identifiability is all from the largest couple of eigenvalues, right?  Can you speak to the significance of this, when it’s the *lowest* eigenvalues that correlated best with validation accuracy in the knowledge transfer task (Table 2)?
- “We propose to complement the notion of data manifolds with the more general one of data foliations” -- how are data foliations more general than data manifolds?
- What is the value of Figs 5 and 8?  Why not show e.g. more traversals, perhaps with the starting point an image from a different dataset?
- Can you speak to computing the DIM in practice?  Could the proposed method easily be applied to an imagenet pre-trained ResNet?
- Is there anything noteworthy about the DIM eigenvalue spectrum on points that the model misclassifies?

---

> ### Author Response · Authors · 2024-11-20
> **Answer to review**
>
> Dear reviewer, we thank you for your corrections and questions. We would like to answer your questions:
> 1. Thank you for pointing out the clash of notation, we have now removed this ambiguous notation for the dataset.
> 2. We have modified our Fig. 4, removing the red points, corresponding to Xor classes and added information, both in the picture and in its caption.
> 3. The phrase "the leaves of a foliation fill the data space" means that, under suitable hypothesis (involutivity in Frobenius theorem), the data space will be the disjoint union of manifolds, called leaves. These manifolds are obtained from the distribution: at each point the distribution is the tangent space to one of such manifolds. The word "it" refers to the distribution D, now we have changed it. If needed, we can add a small appendix clarifying the theory of foliations and manifolds.
> 4. The distribution D at a point p is a 9 dimensional space. So with the phrase "moving tangentially to D", we mean moving on a curve, whose velocity is in such 9 dimensional space. We added few words in the caption of Fig. 2, in the hope to clarify this. The chosen direction is not "random" but from the point 2 to another (picture of a 3) in the dataset. It was chosen solely as an illustration to preview the changes to an images occurring along the path.
> 5. When the distribution is non integrable, for GeLU or Sigmoid activation functions for instance, then we may have a bracket generating distribution. That is, the Lie brackets of its vector fields (see page 4 footnote 2) generate the whole dataspace - or at least a big portion, but Table 1 suggests the former. Whenever a distribution is bracket generating, we can move from any point in the dataspace to any other, by taking directions in D (which is only 9 dimensional). This fact is non obvious, it is a cornerstone of sub Riemannian geometry. In our setting, we are in a different situation: integrability forces us to stay on a single leaf of the data space, hence on a 9 dimensional manifold, if we move along the directions contained in D. This allow us to use geometry on this lower dimensional submanifolds, that would not exists if D wasn't integrable.
> 6. Thank you for pointing out this redundancy. We have modified this accordingly to ease the reading.
> 7. Thank you for your interesting comment. As you mentioned, the training dataset can be identified by averaging several points, and not on a single sample basis. The difference in behavior between the highest and lowest eigenvalues may be due to the exponential decrease (the scale of the plot is logarithmic), so one must be careful when comparing the two.
> 8. A foliation is a collection of submanifolds filling the space (see answer to point 3.). A foliation can thus capture more information than a single submanifold. They are in this sense more general.
> 9. We apologies but we are not sure to understand your question.
> 10. The time cost of computing the DIM is mainly due to the computation of the complete Jacobian matrix of the neural network with respect to its input. This computation is given by automatic differentiation which is already well studied, but might be limiting for very large and complex architectures. However, recall that such automatic differentiation is already used at each step of the training part (wrt to the weight of course, but which are usually higher dimensional than the input space), so if one was able to train a network, one should be able to compute the FIM with a similar time efficiency.
> 11. Thank you for this very interesting question. That is something that we did not look into but we will focus on such data to see what happens. Our first guess is that misclassified points have high uncertainty, and therefore high probability gradient, then DIM's spectrum should be high.
>
> We have highlighted in red the text we modified in the article.

---

### Official Review · Reviewer_Qazy · 2024-11-03

**Soundness:** 3
**Presentation:** 3
**Contribution:** 2
**Rating:** 5
**Confidence:** 3

**Summary:**

The paper introduces a novel data information metric (DIM) adapted from the Fisher information matrix. It then illustrates how to use DIM to visualize the structures of datasets and compute distances between datasets. The paper also explains data foliage structure, which splits data into submanifolds. The foliage structure is further separated into regular and singular points, which gives valuable insights into this dataset. The application is to apply knowledge transfer between *NIST datasets. It transfers knowledge of classification from one dataset to other datasets and obtains good accuracies.

**Strengths:**

1. The foliage structure provides important analysis of datasets, for example, its tangent spaces. Also, the paper visualizes the structures and the eigenvalues comparisons well.

2. The knowledge transfer experiment shows good results and can show the similarity between many datasets This could extend to large datasets and greatly decrease training time. The experiments of moving data in the direction of tangent space also provide a meaningful way to analyze datasets.

3. The paper provides a theoretical background to DIM and its relation with the established results, like the Frobenius theorem and Fisher information matrix.

**Weaknesses:**

1. The experiments are limited to *NIST datasets, so it is not clear whether the knowledge transfer can be extended to more complex datasets and manifolds. For example, the classifier is only fine-tuned with the last layer, and there are no more comparisons. Thus, we cannot conclude whether DIM and knowledge transfer are correlated. If Table 2 has more explanations, that will help the readers to understand the conclusion the authors are trying to get.


2. The theoretical results made assumptions about the neural network architecture, so not applicable to other architectures.

3. Some other papers compute distances between datasets. Reviewer recommends the authors to cite them as well:

Alvarez-Melis, David, and Nicolo Fusi. "Geometric dataset distances via optimal transport." Advances in Neural Information Processing Systems 33 (2020): 21428-21439.

Hua, Xinru, et al. "Dynamic flows on curved space generated by labeled data." Proceedings of the Thirty-Second International Joint Conference on Artificial Intelligence. 2023.

**Questions:**

1. For Figure 4, what are the possible ways to define singular points with ReLU? Or the method does not work with networks with ReLU?

2. How does Figure 6 connect to the knowledge transfer experiments? What is the conclusion drawn from this visualization?

---

> ### Author Response · Authors · 2024-11-20
> **Answer to review**
>
> Dear reviewer, we thank you for your insights and questions. First, we would like to answer your questions:
> 1. In Fig. 4 b) one can see the data foliation for a ReLU network trained on a simple task (mimic the XOR function). We have modified Figure 4 to clarify where the singular and non smooth points are. Our study of singular points is for ReLU only (or any other piecewise linear activation functions), since only then we have a foliation for arbitrary input/output dimensions.
> 2. Since the rank of the DIM and the rank of the Jacobian matrix of the network are the same, one can look at either at the eigenvalues of the first or at the singular values of the second to study the dimension of the distribution. Nonetheless, to clarify the paper, we have replaced Fig. 6 with a plot of the eigenvalues of the DIM so that it is closer to Fig. 7.
>
> Now to comment on the weaknesses you pointed out:
> 1. The limitation to *NIST datasets has been chosen to speed up the computations as it is not the main focus of our paper, but rather a proof of concept of potential applications to the use of the DIM. We would like to point out that, even though CIFAR10 has been scaled to fit the space of MNIST so that they can be compared on transfer knowledge tasks, the CIFAR dataset is quite different from any *NIST.
> 2. The theory developed in this paper is meant as a first step toward the study of more general architectures for future works.
> 3. We thank you for this very relevant suggestion. We have added a citation of these papers to the article in Section 2.
>
> We have highlighted in red the text we modified in the article.

---

### Official Review · Reviewer_375X · 2024-11-04

**Soundness:** 3
**Presentation:** 2
**Contribution:** 3
**Rating:** 5
**Confidence:** 3

**Summary:**

This paper proposes a manifold learning method based on the foliation structure. Particularly, the eigenvalues of the data information matrix are leveraged to measure the difference between two datasets, which reflects the degree of knowledge transfer from one dataset to another. Several benchmark datasets are used to evaluate the performance of knowledge transfer and its relation with the eigenvalues of the data information matrix.

**Strengths:**

1. It is interesting to apply the foliation structure for capturing geometric information of data.

2. The paper constructs a relation between the eigenvalues of the data information matrix and the ability of knowledge transfer between datasets.

**Weaknesses:**

1. Section 1 states that "at least in the cases most relevant for concrete applications, and calls for more sophisticated mathematical modeling". This is too vague to figure out the main point. What kind of concrete applications and challenges are considered here, and what sophisticated mathematical tool is used here for what challenge? More detailed discussions are required to develop this statement.

2. It would be better to re-organize the structure of the submission, especially Section 1. Is it necessary to discuss machine learning and deep learning in Section 1? The connection between this part and the others is unclear.

3. Section 2 is chaotic without a clear structure. In addition, this part just lists some existing works without further insightful discussions.

4. The formal definitions of some mathematical notations are missing. For example, for Eq. (3), it seems that the gradient vectors are used to construct a subspace, while the definition of "span" is missing.

5. The results regarding singular points and the measure are interesting while difficult to understand. It would be helpful to provide some intuitive examples or visualization results, and discuss more about the physical meanings and potential applications.

6. Is it possible to extend the results to other activation functions?

7. The experiment part is weak. Only several simple datasets are used. An interesting connection between DIM eigenvalue and the ability of knowledge transfer is constructed. Based on this, it would be better to apply this observation to the domain adaptation task, which considers how to effectively transfer knowledge between datasets.

**Questions:**

1. Section 1 states that "at least in the cases most relevant for concrete applications, and calls for more sophisticated mathematical modeling". This is too vague to figure out the main point. What kind of concrete applications and challenges are considered here, and what sophisticated mathematical tool is used here for what challenge? More detailed discussions are required to develop this statement.

2. For singular points, it would be helpful to provide some intuitive examples or visualization results, and discuss more about the physical meanings and potential applications.

3. Is it possible to extend the results to other activation functions?

4. It would be better to apply this observation to the domain adaptation task, which considers how to effectively transfer knowledge between datasets.

---

> ### Author Response · Authors · 2024-11-20
> **Answer to review**
>
> Dear reviewer, we thank you for your comments and your questions. First, to answer your questions:
> 1. We reformulated the sentence you mention for more clarity on what modeling and challenges we will consider.
> 2. We have modified our Fig. 4 and its caption, visualizing better the non smooth and singular points of a foliation on the simple XOR example.
> 3. It is possible to extend the results to all piecewise linear activation function (ReLU, LeakyReLU, heavyside,...). We have added a remark after our main result to notify the reader about it.
> 4. Our understanding of the domain adaptation task is that it aims at generalizing to data with the same label as the training dataset. However, our case of study does not need that since we only rely on the fact that all input images (regardless of the dataset they are coming from) are leaving in the same space. That is why we chose the transfer knowledge task to stay more general. As it could lead to very interesting to look at specific cases of domain adaptation tasks in future work, we believe that it would be too much of a change for this paper.
>
> Then, to answer your concerns in the Weaknesses section:
> 1. See answer to question 1.
> 2. We reorganized and shortened the aforementioned paragraph in Section 1.
> 3. We gave more structure to Section 2 and added a introduction sentence.
> 4. We have added a footnote for the missing definition of the "span". If needed, we can add a small appendix clarifying the theory of foliations and manifolds.
> 5. As stated in our answer to question 2, we have modified our Fig. 4 so that it helps the reader to understand better our theoretical result on a simple case. Though, we are not sure to see what physical meaning you are talking about in your comment.
> 6. See answer to question 3.
> 7. See answer to question 4.
>
> We have highlighted in red the text we modified in the article.

---

### Meta-Review · Area_Chair_qf1C · 2024-12-20

**Metareview:**

This paper proposes an information geometry analysis for the analysis of data manifold learning based on the notion of foliation structure. It proposes to consider “data information matrix” (DIM), a variation of the Fisher information matrix to detect a foliation structure on the data space. The eigenvalues of this data information matrix can be used to measure the difference between two datasets, which reflects the degree of knowledge transfer from one dataset to another. The paper provides theoretical results and an empirical evaluation.

Strengths:
-combination of differentiable geometry and information geometry,
-using foliation structure is novel in this context,
-the theoretical framework is solid.

Weaknesses:
- the paper needs a restructuration,
-some parts are difficult to understand and some precisions are needed,
-experimental evaluation is weak.

All the reviewers found that the paper has some merits but lacks enough clarity and a convincing experimental study.
During rebuttal, authors have given answers to the remarks of the reviewers by providing additional explanations and precisions. But they did not report additional experimental results.
During the discussion, reviewers recognized that the paper provides an interesting approach but they agreed on the fact that the contribution is not clear enough and the experiments are not convincing.

As a consequence I propose rejection.
However, this paper has an interesting basis and I encourage authors to improve their work for other venues.

**Additional Comments On Reviewer Discussion:**

All the reviewers rated the paper as below the acceptance bar (score 5).

During the discussion, reviewer L83N mentioned that the revision did not change his opinion: The paper has merits but lacks clarity, and it did not change materially during the revision process.

Reviewer qT62 indicated that he offered constructive suggestions for this paper, hoping the authors would present new experiments. Authors did not respond to his suggestions and did not provide convincing alternatives. He indicated that he has to maintain his score in this context.

Other reviewers did not support the paper.

Rejection was then proposed.

---

### Decision · Program_Chairs · 2025-01-22

Reject